# Robust Amino-Functionalized Mesoporous Silica Hollow Spheres Templated by CO_2_ Bubbles

**DOI:** 10.3390/molecules27010053

**Published:** 2021-12-22

**Authors:** Hongjuan Wang, Xuefei Liu, Olena Saliy, Wei Hu, Jingui Wang

**Affiliations:** 1School of Chemistry and Chemical Engineering, Qilu University of Technology (Shandong Academy of Sciences), Jinan 250353, China; hongjuanwang2015@163.com (H.W.); paowuxian124@163.com (X.L.); 2Kyiv College, Qilu University of Technology (Shandong Academy of Sciences), Jinan 250353, China; saliy.oo@knutd.edu.ua; 3Department of Industrial Pharmacy, Faculty of Chemical and Biopharmaceutical Technologies, Kyiv National University of Technologies and Design, 01011 Kyiv, Ukraine

**Keywords:** hollow, mesoporous silica, amino-functionalized, bubble template, CO_2_ capture, drug-controlled release

## Abstract

Hollow-structured mesoporous silica has wide applications in catalysis and drug delivery due to its high surface area, large hollow space, and short diffusion mesochannels. However, the synthesis of hollow structures usually requires sacrificial templates, leading to increased production costs and environmental problems. Here, for the first time, amino-functionalized mesoporous silica hollow spheres were synthesized by using CO_2_ gaseous bubbles as templates. The assembly of anionic surfactants, co-structure directing agents, and inorganic silica precursors around CO_2_ bubbles formed the mesoporous silica shells. The hollow silica spheres, 200–400 nm in size with 20–30 nm spherical shell thickness, had abundant amine groups on the surface of the mesopores, indicating excellent applications for CO_2_ capture, Knoevenagel condensation reaction, and the controlled release of Drugs.

## 1. Introduction

Since mesoporous materials were discovered in the 1990s [1,2], they have been known for their high specific surface areas, large pore volumes, and continuously adjustable mesoporous channels, which enables them to be widely used in the fields of catalysis, separation, and biomedicine [3,4,5,6,7]. Hollow-structured mesoporous silica shows better application prospects for catalysis and drug delivery, attributable to its large hollow space, low density, and short diffusion mesoporous channels. 

Various methods for preparing mesoporous materials with hollow structures have been developed. Many nanomaterials have been employed as templates to construct hollow structures, including vesicles [8,9,10], bubbles [7,11,12,13], microemulsions [14,15,16,17,18], metals, metal oxides, semiconductor materials, polymer microspheres [19,20], and other materials [21]. However, most of the templates were sacrificial and required a removal process to create a hollow cavity, which led to many problems such as complicated preparation processes, high preparation costs, and environmental pollution. Of these various methods, gaseous bubbles were attractive as templates, due to their low cost and environmental friendliness. However, the direct wrapping of gaseous bubbles was not easy, as the bubbles were always dynamic.

In this study, hollow-structured amino-functionalized mesoporous silica was prepared using CO_2_ bubbles as templates. An assembly of anionic surfactant, a silane coupling agent, and inorganic silica precursors around CO_2_ bubbles formed the mesoporous silica shells, generated by high-pressure CO_2_. The CO_2_ bubbles were used as templates for hollow structures, and the anionic surfactants were used as templates for mesopores. By this method, amino-functionalized mesoporous silica hollow spheres could be one-pot prepared, in contrast with other methods, such as the cationic surfactant method [11].

The synthesized hollow mesoporous silica was robust. As the hollow templates of CO_2_ bubbles did not require further treatment for removal, the method was economical and environmentally friendly, compared to other sacrificial templates that required removal by calcination [19]. The obtained hollow mesoporous silica showed good amounts of CO_2_ and high performance in Knoevenagel reaction, due to the presence of abundant amine groups. In addition, mesoporous silica hollow spheres displayed excellent performance in drug-controlled release.

## 2. Results and Discussion

As shown in Figure 1a,b, when the pressure of CO_2_ gas was 1.0 MPa during synthesis, the obtained sample had a hollow structure and a smooth spherical surface. The hollow sphere was 200–400 nm in size. The shell thickness was 20–30 nm. There were numerous mesopores in the shell. Some hollow spheres were composed of several smaller hollow spheres, comprised of only one cavity without barriers at the junction. This was because the CO_2_ bubble template was always dynamic and unstable during the wrapping process. The phenomenon of bubble-merging was also evident in the bubble template. If the CO_2_ pressure were lower, disordered amorphous silica nanoparticles and/or solid mesoporous silica particles would be formed (Figure 1c,d). This indicated that the wrapping of a dynamic gas bubble by mesoporous silica should be precisely controlled.

Figure 2a shows that the nitrogen adsorption-desorption isotherms of the obtained amino-functionalized mesoporous silica hollow spheres yielded type-IV isotherms with an adsorption step at relative pressures between 0.4 and 0.8, due to the capillary condensation of the filling nitrogen in the mesopores. The sharp increase of adsorption amounts of nitrogen at the higher relative pressure (>0.95) indicated the small size of the synthesized mesoporous silica, which was consistent with the results of TEM testing. The pore-size distribution curve (inset of Figure 1a) showed that the obtained amino-functionalized mesoporous silica hollow spheres had a single peak centered at 4.0 nm. The BET specific surface area was 361 m^2^·g^−1^, and the total pore volume was 0.95 cm^3^·g^−1^. The presence of a hysteresis loop at relative pressures of 0.45–0.95 indicated a hollow structure with mesoporous walls. In addition, the presence of amine groups was observed in the FTIR spectrum (Appendix A), and the content of amine moiety was 1.13 mmol·g^−1^.

Based on the hollow structure and amino-functionalized characteristics of the synthesized material, it was applied to CO_2_ capture, Knoevenagel reaction, and MPT drug-controlled release testing. Figure 2b shows that the CO_2_ adsorption curve had an upward trend throughout the process, and gradually trended to equilibrium at 60 min. The maximal adsorption amount could reach 0.69 mmol·g^−1^, suggesting that each amine could capture approximately 0.6 CO_2_ molecules. This implied a certain physical adsorption in addition to chemical adsorption, based on the quantitative reaction in which one carbon dioxide molecule required two amines.

Amino-functionalized mesoporous silica has been shown to have catalytic activity in Knoevenagel reactions. Amine moieties were present on the inner surface of the mesochannels. The more the amine was exposed, the better the catalytic activity [22]. As shown in Figure 2c, the conversion gradually increased with time, reaching about 80% conversion within 1 h, and about 100% at 4 h. This is better than traditional amino-functionalized MCM-41 (Appendix A), indicating a good catalyst for Knoevenagel reaction due to the abundant exposed amine groups and the short mesopores in the thin shell. This was beneficial for reactant transport to contact the catalytic active sites of amines.

Figure 2d shows the MPB drug-release curve. Size distribution was measured by DLS before and after adding metoprolol to the spheres (Appendix A), showing no obvious changes in size after adding metoprolol to the spheres. This implied that there was no obvious aggregation after introducing MPB. The drug-release amount could reach 50% within 1 h and 90% within 5 h, indicating that the synthesized mesoporous hollow sphere could achieve controlled drug release. which demonstrated potential for stimulus-response release and targeted therapy.

## 3. Materials and Methods

### 3.1. Synthesis of Amino-Functionalized Mesoporous Silica Hollow Spheres

All chemicals were purchased from Shanghai Macklin Biochemical Co., Ltd. (Shanghai, China) and used as received. In a typical synthesis, 1.0 mmol of N-Lauroylsarcosine sodium (as anionic surfactant) was completely dissolved in 30 mL of deionized water at room temperature. Next, a mixture of 1.5 mL ethyl orthosilicate (an inorganic silica source) and 0.12 mL 3-aminopropyltriethoxysilane (a co-structure directing agent) was added under vigorous stirring. The total solution was transferred to a Tetrafluoroethylene-lined stainless steel autoclave. Next, CO_2_ gas was introduced at a pressure of 1.0 MPa. The mixture was stirred at room temperature for 24 h, then placed in an oven at 80 °C for another 4 h. The resulting product was extracted with 10 wt% hydrochloric acid in acetonitrile (the ratio of solid production to solution was 1 g:20 mL) at room temperature for 24 h to remove surfactant, to expose the amino groups on the surface of the mesopores. The final product was washed/exchanged with 1.0 wt% ammonia solution (the ratio of solid production to solution was 1 g:20 mL), filtered, washed with distilled water, and dried overnight at 80 °C.

### 3.2. Carbon Dioxide Capture Experiment

The CO_2_ performance test was carried out in a thermogravimetric analyzer (TA SDT Q600, New Castle, DE, USA). In the pretreatment stage, 5.0 mg of the sample was placed into a small crucible and heated at temperatures ranging from room temperature to 150 °C, at a heating rate of 10 °C/min in a nitrogen atmosphere for 30 min. When the sample naturally cooled to 30 °C and was kept at this temperature, pure CO_2_ was introduced at 100 mL·min^−1^ for 60 min. The change in mass was recorded by the instrument.

### 3.3. Knoevenagel Reaction

50 mg of the catalyst was placed in a reaction vessel, and 1.4 mL of toluene, 0.02 g of naphthalene (internal standard), 133 μL of benzaldehyde, and 136 μL of ethyl cyanoacetate were added. The mixture was heated at 30 °C in an oil bath under stirring. Small amounts of reaction mixture were removed at different times and analyzed by Shimadzu GC-2014 gas chromatography (Kyoto, Japan) equipped with a 30 m TC-1 capillary column and a flame ionization detector).

### 3.4. Metoprolol Tartrate (MPT) Drug-loading and Release Experiments

5 mg of MPT was fully dissolved in 2 mL of ethanol. Next, 50 mg hollow silica spheres were added and dispersed under sonication. The mixture was placed in a 50 °C oven to evaporate ethanol for 24 h. Subsequently, 2 mL of ethanol was used to wash the unabsorbed drug on the surface of the hollow sphere. The resulting mixture was centrifuged and dried at 50 °C.

Next, drug-loaded mesoporous silica hollow spheres were used for drug-release experiments. The release process was based on the study reported at [23]. Ten mg of the sample was added to 100 mL of phosphate buffer (pH = 7.4), with a stirring speed of 100 rpm at room temperature, and 2 mL of extraction solution was removed each time. The absorbance of the sample was measured with a Persee TU-1901 UV-Vis spectrophotometer (Beijing, China) at a wavelength of 274 nm.

### 3.5. Characterization

A TEM image was obtained by a JEOL JEM-2100 TEM microscope (Tokyo, Japan) operating at 200 kV. Nitrogen adsorption isotherms were conducted by a Micromeritics TrisStar II 3020 sorption analyzer (Norcross, GA, USA) at 77 K. The specific surface area and pore volume were analyzed by the BET (Brunauer-Emmett-Teller) method, and the pore-size distribution was analyzed from the adsorption data using BJH (Barett-Joyner-Halenda). CO_2_ adsorption was tested by a TA SDT Q600 simultaneous thermal analyzer (New Castle, DE, USA). The flow rate of CO_2_ was set to 100 mL·min^−1^. The nitrogen elemental content was evaluated using an Elementar Vario EL Cube elemental analyzer (Langenselbold, Germany). Particle size distribution was tested on Malvern Zetasizer Nano ZS90 equipment (Malvern, England). Fourier Transform Infrared (FTIR) spectra were measured on a Thermo Scientific Nicolet iS50 spectrometer (Madison, WI, USA).

## 4. Conclusions

Amino-functionalized hollow mesoporous silica, 200–400 nm in size with 20–30 nm shell thickness, was prepared by using carbon dioxide bubbles as a template. The prepared hollow spheres possessed 1.13 mmol·g^−1^ amine moiety, which could capture 0.69 mmol·g^−1^ CO_2_ molecules within 60 min. In addition, the robust hollow mesoporous silica could be well-dispersed in aqueous or organic systems (Appendix A, Appendix A), showing good catalytic activity in Knoevenagel reaction and excellent MPB drug-controlled release. This implies potential application in CO_2_ capture from air and in stimulus-response release for targeted therapy.

## Figures and Tables

**Figure 1 molecules-27-00053-f001:**
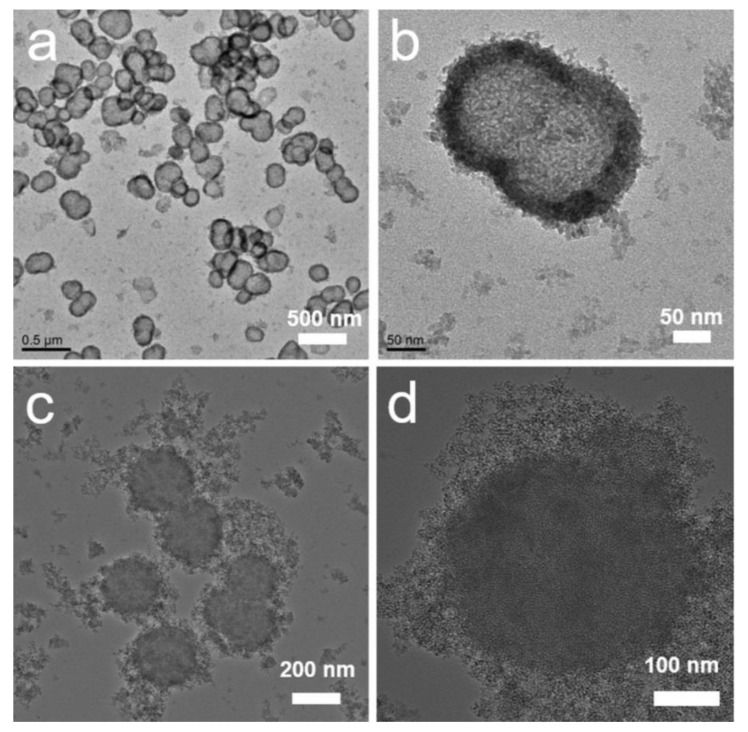
TEM images of the amino-functionalized mesoporous silica synthesized under different CO_2_ pressures: (**a**,**b**) 1.0 MPa; (**c**,**d**) 0.2 MPa.

**Figure 2 molecules-27-00053-f002:**
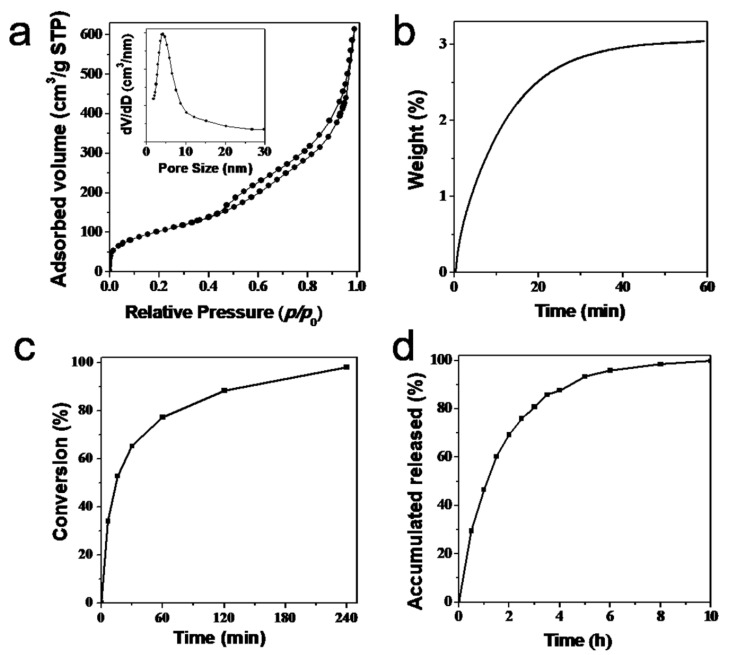
(**a**) Nitrogen adsorption-desorption isotherms and (inset) the pore-size distribution curves; (**b**) CO_2_ adsorption curve; (**c**) Knoevenagel reaction curve; and (**d**) metoprolol tartrate (MPT) drug release of amino-functionalized mesoporous silica hollow spheres.

## Data Availability

Not Applicable.

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
