# Peer review of "Robust Amino-Functionalized Mesoporous Silica Hollow Spheres Templated by CO2 Bubbles"

_molecules, 2021, doi:10.3390/molecules27010053_

Round 1

Reviewer 1 Report

This manuscript presents the synthesis and characterization of amino-functionalized mesoporous silica hollow spheres. Even though the authors highlighted their CO2 bubble templating strategy, there are already reported (one-step) routes to the amine-functionalized hollow mesoporous silica particles. Nevertheless, they have shown interesting properties for future practical applications. Hence, I would recommend publication in this journal after dealing with the minor issues written below.

1. What is the distinct advantage of using CO2 at high pressures like 10 bar? If there are two options with and without using high-pressure conditions, I will choose not to use the 10-bar reaction. Moreover, the authors displayed SEM images with different magnifications for the two samples from 2 and 10 bar. This made it difficult to compare the samples in a fair manner.

2. It is good to know that the synthesized, amine-functionalized hollow silica exhibited ordered mesoporosity, CO2 adsorption capacity, base catalytic activity, and drug release function. However, there is no information about the controllability of amine quantity and/or porous textures, since the authors only showed the single point of the synthesis. Is it possible to tune these properties?

3. Regarding the three applications, how close (or far) is the material to the current industry standard? In other words, the authors should have compared with other hollow silica and/or state-of-the-art materials at least in one representative application.

4. There is no characterization for proving the presence of amine functional groups in the material. The authors should add characterization results to use the term “amino-functionalized” in the title and the abstract of this work.

5. Please add citations: Small 2016, 12, No. 37, 5169–5177 /
Ind. Eng. Chem. Res. 2020, 59, 723−731 /
Chem. Commun., 2009, 2365–2367 /
ACS Appl. Mater. Interfaces 2015, 7, 1040−1045 /
Nature 575, 40-41 (2019)

Reviewer 2 Report

English in the abstract and elsewhere
    it is readable and unambiguous, but some sentences could be rewritten
    Sentences 42-44 do not even have a verb

64 the wrapping of a dynamic gas bubble by mesoporous silica should be precisely    controlled
        What does it mean, how to control individual bubbles?
What do we know about the behavior and size distribution of the introduced bubbles? 

Figure 2
    C and D if it has no deviations, it was one run?
    C to which reactant do they have a defined conversion?
There is not much in Materials and Methods for that analysis either

91 "So, the more the amine exposed, the better the catalytic activity was."
        Is that a general consideration? An observation from the literature? Or are you comparing different experiments?
93 No mention of a negative control

95 beneficial to the reactant transport to contact the catalytic active sites of amines
        That makes sense, but is there any comparison? 

99 stimulus-response release
        Is there an external stimulus here, or just simple diffusion out?

112 a little more detail would be useful on the synthesis procedure, for example what was the volume of ammonia solution to extract

115 what type of TG analyser? are we sure that CO2 can leave freely and not create overpressure?
116 the 30 min is then at constant temperature?
120 When you say "Typical", how many repetitions were there?
        I think the reaction deserves an equation with the structure of the products, even though this is a material science article
125 what peak did they observe in the GC? what came out of the internal standard?
        Negative control, how did it run without catalyst?
128 again: how many repeats?
134 phosphate buffer, what concentration, pH?
139 "TEM image of the product is characterized" doesn't really make sense
149 1.13 mmol-g-1 amine moiety
            does it have any variation? is it homogeneous?
151 well-dispersed in aqueous systems:
            where do they have an example of an aqueous system? just ethanol, right?

I didn't find any total synthesis yield.

Translated with www.DeepL.com/Translator (free version)

Reviewer 3 Report

In this manuscript, the authors reported for the first time amino-functionalized mesoporous silica hollow spheres synthesized by using CO2 gaseous bubbles as templates. The paper lacks a sort of novelty and originality that fit for a paper being considered publication. Therefore, the authors should improve the manuscript to show the importance of the topic. Innovation is not outstanding enough. It is necessary to highlight the innovation to increase the novelty of the article. The detailed comments are as follows:

  1. The authors may need to briefly address the difference(s) between the current manuscript and other similar published review articles in the Introduction section.
  1. Authors need to work on the presentation of the manuscript, writing and the discussion of the results, otherwise I would not recommend this article for publication.  
  2. Overall, the hollow spheres should be more characterized, including FTIR results, XRD and thermogravimetric analysis. For instance, I suggest the authors to measure zeta potential values of the materials as function of pH and also DLS measurements. DLS measured size distribution must be provided before and after adding metoprolol to the spheres in order to explore the possibility of particles aggregation.
  3. The release process modeling using e.g. Weibull model should be provided.
  4. In the conclusions, the authors should also provide an outlook of the challenges and potential future directions.
  5. I encourage authors to keep working on the experimental performance of the material before publishing it

Round 2

Reviewer 3 Report

The authors responded to the comments. Manuscript was improved in accordance to my suggestions and I have no further objection to this paper.